# UneVEn: Universal Value Exploration for Multi-Agent Reinforcement Learning

## Abstract

This paper focuses on cooperative value-based multi-agent reinforcement learning (MARL) in the paradigm of centralized training with decentralized execution (CTDE). Current state-of-the-art value-based MARL methods leverage CTDE to learn a centralized joint-action value function as a monotonic mixing of each agent's utility function, which enables easy decentralization. However, this monotonic restriction leads to inefficient exploration in tasks with *nonmonotonic* returns due to suboptimal approximations of the values of joint actions. To address this, we present a *novel* MARL approach called Universal Value Exploration (UneVEn), which uses universal successor features (USFs) to learn policies of tasks *related* to the target task, but with simpler reward functions in a sample efficient manner. UneVEn uses novel action-selection schemes between randomly sampled related tasks during exploration, which enables the monotonic joint-action value function of the target task to place more importance on useful joint actions. Empirical results on a challenging cooperative predator-prey task requiring significant coordination amongst agents show that UneVEn significantly outperforms state-of-the-art baselines.

## 1 Introduction

Learning control policies for cooperative multi-agent reinforcement learning (MARL) remains challenging as agents must search the joint-action space, which grows exponentially with the number of agents. Current state-of-the-art value-based methods such as VDN (Sunehag et al., 2017) and QMIX (Rashid et al., 2020b) learn a *centralized* joint-action value function as a *monotonic* factorization of *decentralized* agent utility functions and can therefore cope with large joint action spaces. Due to this monotonic factorization, the joint-action value function can be decentrally maximized as each agent can simply select the action that maximizes its corresponding utility function.

This monotonic restriction, however, prevents VDN and QMIX from representing *nonmonotonic* joint-action value functions (Mahajan et al., 2019) where an agent's best action depends on what actions the other agents choose. For example, consider a predator-prey task where *at least three* agents need to coordinate to capture a prey and any capture attempts by fewer agents are penalized with a penalty of magnitude $p$. As a result, both VDN and QMIX tend to get stuck in a suboptimal equilibrium (also called the *relative overgeneralization* pathology, Panait et al., 2006; Wei et al., 2018) in which agents simply avoid the prey (Mahajan et al., 2019; Böhmer et al., 2020). This happens for two reasons. First, depending on $p$, successful coordination by at least three agents is a needle in the haystack and any step towards it is penalized. Second, the monotonically factorized joint-action value function lacks the representational capacity to distinguish the values of coordinated and uncoordinated joint actions during exploration.

Recent work addresses the problem of inefficient exploration by VDN and QMIX due to monotonic factorization. QTRAN (Son et al., 2019) and WQMIX (Rashid et al., 2020a) address this problem by weighing *important joint actions* differently, which can be found by simultaneously learning a centralized value function, but these approaches still rely on inefficient $\epsilon$-greedy exploration which may fail on harder tasks (e.g., the predator-prey task above with higher value of $p$). MAVEN (Mahajan et al., 2019) learns an ensemble of monotonic joint-action value functions through committed exploration by maximizing the entropy of the trajectories conditioned on a latent variable. Their exploration focuses on diversity in the joint team behaviour using mutual information. By contrast,

this paper proposes Universal Value Exploration (UneVEn), which follows the intuitive premise that tasks with a simpler reward function than the *target* task (e.g., a smaller miscoordination penalty in predator-prey) can be efficiently solved using a monotonic factorization of the joint-action value function. Therefore, UneVEn samples tasks *related* to the target task, that are often easier to solve, but often have *similar* important joint actions. Selecting actions based on these related tasks during exploration can bias the monotonic approximation of the value function towards important joint actions of the target task (Son et al., 2019; Rashid et al., 2020a), which can overcome relative overgeneralization. To leverage the policies of the sampled related tasks, which only differ in their reward functions, UneVEn uses Universal Successor Features (USFs, Borsa et al., 2018) which have demonstrated excellent zero-shot generalization in single-agent tasks with different reward functions (Barreto et al., 2017; 2020). USFs generalize policy dynamics over tasks using Universal Value Functions (UVFs, Schaul et al., 2015), along with Generalized Policy Improvement (GPI, Barreto et al., 2017), which combines solutions of previous tasks into new policies for unseen tasks.

Our contributions are as follows. First, we propose Multi-Agent Universal Successor Features (MAUSFs) factorized into novel decentralized agent-specific SFs with value decomposition networks (Sunehag et al., 2017) from MARL. This factorization enables agents to compute decentralized greedy policies and to perform decentralized local GPI, which is particularly well suited for MARL, as it allows to maximize over a combinatorial set of agent policies. Second, we propose Universal Value Exploration (UneVEn), which uses novel *action-selection* schemes based on *related tasks* to solve tasks with nonmonotonic values with monotonic approximations thereof. We evaluate our novel approach in predator-prey tasks that require significant coordination amongst agents and highlight the relative overgeneralization pathology. We empirically show that UneVEn with MAUSFs significantly outperforms current state-of-the-art value-based methods on the target tasks and in zero-shot generalization (Borsa et al., 2018) across MARL tasks with different reward functions, which enables us to leverage UneVEn effectively.

## 2 BACKGROUND

**Dec-POMDP**: A fully cooperative decentralized multi-agent task can be formalized as a *decentralized partially observable Markov decision process* (Dec-POMDP, Oliehoek et al., 2016) consisting of a tuple $G = \langle \mathcal{S}, \mathcal{U}, P, R, \Omega, O, n, \gamma \rangle$. $s \in \mathcal{S}$ describes the true state of the environment. At each time step, each agent $a \in \mathcal{A} \equiv \{1, ..., n\}$ chooses an action $u^a \in \mathcal{U}$, forming a joint action $\boldsymbol{u} \in \boldsymbol{\mathcal{U}} \equiv \mathcal{U}^n$. This causes a transition in the environment according to the state transition kernel $P(s'|s, \boldsymbol{u}) : \mathcal{S} \times \boldsymbol{\mathcal{U}} \times \mathcal{S} \rightarrow [0, 1]$. All agents are collaborative and share therefore the same reward function $R(s, \boldsymbol{u}) : \mathcal{S} \times \boldsymbol{\mathcal{U}} \rightarrow \mathbb{R}$ and $\gamma \in [0, 1)$ is a discount factor.

Due to *partial observability*, each agent $a$ cannot observe the true state $s$, but receives an observation $o^a \in \Omega$ drawn from observation kernel $o^a \sim O(s, a)$. At time $t$, each agent $a$ has access to its action-observation history $\tau_t^a \in \mathcal{T}_t \equiv (\Omega \times \mathcal{U})^t \times \Omega$, on which it conditions a stochastic policy $\pi^a(u_t^a|\tau_t^a)$. $\boldsymbol{\tau}_t \in \mathcal{T}_t^n$ denotes the histories of all agents. The joint stochastic policy $\boldsymbol{\pi}(\boldsymbol{u}_t|s_t, \boldsymbol{\tau}_t) \equiv \prod_{a=1}^n \pi^a(u_t^a|\tau_t^a)$ induces a joint-action value function : $Q^\pi(s_t, \boldsymbol{\tau}_t, \boldsymbol{u}_t) = \mathbb{E}[G_t|s_t, \boldsymbol{\tau}_t, \boldsymbol{u}_t]$, where $G_t = \sum_{i=0}^\infty \gamma^i r_{t+i}$ is the *discounted return*.

**CTDE**: We adopt the framework of *centralized training and decentralized execution* (CTDE Kraemer & Banerjee, 2016), which assumes access to all action-observation histories $\boldsymbol{\tau}_t$ and global state $s_t$ during training, but each agent's decentralized policy $\pi^a$ can only condition on its own action-observation history $\tau^a$. This approach can exploit information that is not available during execution and also freely share parameters and gradients, which improves the sample efficiency considerably (see e.g., Foerster et al., 2018; Rashid et al., 2020b; Böhmer et al., 2020).

**Value Decomposition Networks**: A naive way to learn in MARL is *independent Q-learning* (IQL, Tan, 1993), which learns an independent action value function $Q^a(\tau_t^a, u_t^a; \theta^a)$ for each agent $a$ that conditions only on its local action-observation history $\tau_t^a$. To make better use of other agents' information in CTDE, *value decomposition networks* (VDN, Sunehag et al., 2017) represent the joint-action value function $Q_{tot}$ as a sum of per-agent *utility functions* $Q^a$: $Q_{tot}(\boldsymbol{\tau}, \boldsymbol{u}; \theta) \equiv \sum_{a=1}^n Q^a(\tau^a, u^a; \theta)$. Each $Q^a$ still conditions only on individual action-observation histories and can be represented by an agent network that shares parameters across all agents. The joint-action value function $Q_{tot}$ can be trained using Deep Q-Networks (DQN, Mnih et al., 2015). Compared to VDN, QMIX (Rashid et al., 2020b) allows joint-action value function $Q_{tot}$ to be represented as

a nonlinear *monotonic* combination of individual utility functions. The greedy joint action in both VDN and QMIX can be computed decentrally by individually maximizing each agent's utility. See OroojlooyJadid & Hajinezhad (2019) for a more in-depth overview of cooperative deep MARL.

**Task based Universal Value Functions**: In this paper, we consider tasks that differ only in their reward functions $R_{\boldsymbol{w}}(s, \boldsymbol{u}) \equiv \boldsymbol{w}^\top \boldsymbol{\phi}(s, \boldsymbol{u})$, which are linear combinations of a set of basis functions $\boldsymbol{\phi} : \mathcal{S} \times \mathcal{U} \rightarrow \mathbb{R}^d$. Intuitively, the basis functions $\boldsymbol{\phi}$ encode potentially rewarded events, such as opening a door or picking up an object. We use the weight vector $\boldsymbol{w}$ to denote the task with reward function $R_{\boldsymbol{w}}$. Universal Value Functions (UVFs, Schaul et al., 2015) is an extension of DQN that learns a *generalizable* value function conditioned on tasks. UVFs are typically of the form $Q^\pi(s_t, \boldsymbol{u}_t, \boldsymbol{w})$ to indicate the action-value function of task $\boldsymbol{w}$ under policy $\pi$ at time $t$ as:

$$Q^\pi(s_t, \boldsymbol{u}_t, \boldsymbol{w}) = \mathbb{E}^\pi \big[ \sum_{i=0}^\infty \gamma^i R_{\boldsymbol{w}}(s_{t+i}, \boldsymbol{u}_{t+i}) \, \big| \, s_t, \boldsymbol{u}_t \big] = \mathbb{E}^\pi \big[ \sum_{i=0}^\infty \gamma^i \boldsymbol{\phi}(s_{t+i}, \boldsymbol{u}_{t+i})^\top \boldsymbol{w} \, \big| \, s_t, \boldsymbol{u}_t \big]. \quad (1)$$

**Successor Features**: The Successor Representation (Dayan, 1993) has been widely used in single-agent settings (Barreto et al., 2017; 2018; Borsa et al., 2018) to generalize across tasks with given reward specifications. By simply rewriting the definition of the action value function $Q^\pi(s_t, \boldsymbol{u}_t, \boldsymbol{w})$ of task $\boldsymbol{w}$ from Equation 1 we have:

$$Q^\pi(s_t, \boldsymbol{u}_t, \boldsymbol{w}) \;\; = \;\; \mathbb{E}^\pi \big[ \sum_{i=0}^\infty \gamma^i \boldsymbol{\phi}(s_{t+i}, \boldsymbol{u}_{t+i}) \, \big| \, s_t, \boldsymbol{u}_t \big]^\top \boldsymbol{w} \;\; \equiv \;\; \boldsymbol{\psi}^\pi(s_t, \boldsymbol{u}_t)^\top \boldsymbol{w} \,, \quad (2)$$

where $\boldsymbol{\psi}^\pi(s, \boldsymbol{u})$ are the Successor Features (SFs) under policy $\pi$. For the optimal policy $\pi_{\boldsymbol{z}}^\star$ of task $\boldsymbol{z}$, the SFs $\boldsymbol{\psi}^{\pi_{\boldsymbol{z}}^\star}$ summarize the dynamics under this policy, which can then be weighted with any reward vector $\boldsymbol{w} \in \mathbb{R}^d$ to instantly evaluate policy $\pi_{\boldsymbol{z}}^\star$ on it: $Q^{\pi_{\boldsymbol{z}}^\star}(s, \boldsymbol{u}, \boldsymbol{w}) = \boldsymbol{\psi}^{\pi_{\boldsymbol{z}}^\star}(s, \boldsymbol{u})^\top \boldsymbol{w}$.

**Universal Successor Features and Generalized Policy Improvement**: Borsa et al. (2018) introduce universal successor features (USFs) which learns SFs conditioned on tasks using the *generalization* power of UVFs. Specifically, they define UVFs of the form $Q(s, \boldsymbol{u}, \boldsymbol{z}, \boldsymbol{w})$ which represents the value function of policy $\pi_{\boldsymbol{z}}$ evaluated on task $\boldsymbol{w} \in \mathbb{R}^d$. These UVFs can be factored using the SFs property (Equation 2) as: $Q(s, \boldsymbol{u}, \boldsymbol{z}, \boldsymbol{w}) = \boldsymbol{\psi}(s, \boldsymbol{u}, \boldsymbol{z})^\top \boldsymbol{w}$, where $\boldsymbol{\psi}(s, \boldsymbol{u}, \boldsymbol{z})$ are the USFs that generate the SFs induced by task-specific policy $\pi_{\boldsymbol{z}}$. One major advantage of using SFs is the ability to *efficiently* do generalized policy improvement (GPI, Barreto et al., 2017), which allows a new policy to be computed for *any unseen* task based on instant policy evaluation of a *set* of policies on that unseen task with a simple dot-product. Formally, given a set $\mathcal{C} \subseteq \mathbb{R}^d$ of tasks and their corresponding SFs $\{\boldsymbol{\psi}(s, \boldsymbol{u}, \boldsymbol{z})\}_{\boldsymbol{z} \in \mathcal{C}}$ induced by corresponding policies $\{\pi_{\boldsymbol{z}}\}_{\boldsymbol{z} \in \mathcal{C}}$, a new policy $\pi_{\boldsymbol{w}}'$ for any unseen task $\boldsymbol{w} \in \mathbb{R}^d$ can be derived using:

$$\pi_{\boldsymbol{w}}'(s) \;\; \in \;\; \arg\max_{\boldsymbol{u} \in \mathcal{U}} \max_{\boldsymbol{z} \in \mathcal{C}} Q(s, \boldsymbol{u}, \boldsymbol{z}, \boldsymbol{w}) \;\; = \;\; \arg\max_{\boldsymbol{u} \in \mathcal{U}} \max_{\boldsymbol{z} \in \mathcal{C}} \boldsymbol{\psi}(s, \boldsymbol{u}, \boldsymbol{z})^\top \boldsymbol{w}. \quad (3)$$

Setting $\mathcal{C} = \{\boldsymbol{w}\}$ allows us to revert back to UVFs, as we evaluate SFs induced by policy $\pi_{\boldsymbol{w}}$ on task $\boldsymbol{w}$ itself. However, we can use any set of tasks that are similar to $\boldsymbol{w}$ based on some similarity distribution $\mathcal{D}(\cdot | \boldsymbol{w})$. The computed policy $\pi_{\boldsymbol{w}}'$ is guaranteed to perform no worse on task $\boldsymbol{w}$ than *each* of the policies $\{\pi_{\boldsymbol{z}}\}_{\boldsymbol{z} \in \mathcal{C}}$ (Barreto et al., 2017), but often performs much better. SFs thus enable efficient use of GPI, which allows *reuse* of learned knowledge for zero-shot generalization.

## 3 MULTI-AGENT UNIVERSAL SUCCESSOR FEATURES

In this section, we introduce Multi-Agent Universal Successor Features (MAUSFs), extending single-agent USFs (Borsa et al., 2018) to multi-agent settings and show how we can learn generalized *decentralized* greedy policies for agents. The USFs based centralized joint-action value function $Q_{tot}(\boldsymbol{\tau}, \boldsymbol{u}, \boldsymbol{z}, \boldsymbol{w})$ allows evaluation of joint policy $\boldsymbol{\pi}_{\boldsymbol{z}} = \langle \pi_{\boldsymbol{z}}^1, \ldots, \pi_{\boldsymbol{z}}^n \rangle$ comprised of local agent policies $\pi_{\boldsymbol{z}}^a$ of *same* task $\boldsymbol{z}$ on task $\boldsymbol{w}$. However, each agent $a$ may execute a different policy $\pi_{\boldsymbol{z}^a}^a$ of different task $\boldsymbol{z}^a \in \mathcal{C}$, resulting in a combinatorial set of joint-policies. Maximizing over all combinations $\bar{\boldsymbol{z}} \equiv \langle \boldsymbol{z}^1, \ldots, \boldsymbol{z}^n \rangle \in \mathcal{C}^n$ should therefore enormously improve GPI. To enable this flexibility, we define joint-action value function ($Q_{tot}$) of joint policy $\boldsymbol{\pi}_{\bar{\boldsymbol{z}}} = \{\pi_{\boldsymbol{z}^a}^a\}_{\boldsymbol{z}^a \in \mathcal{C}}$ evaluated on any task $\boldsymbol{w} \in \mathbb{R}^d$ as: $Q_{tot}(\boldsymbol{\tau}, \boldsymbol{u}, \bar{\boldsymbol{z}}, \boldsymbol{w}) = \boldsymbol{\psi}_{tot}(\boldsymbol{\tau}, \boldsymbol{u}, \bar{\boldsymbol{z}})^\top \boldsymbol{w}$, where $\boldsymbol{\psi}_{tot}(\boldsymbol{\tau}, \boldsymbol{u}, \bar{\boldsymbol{z}})$ are the MAUSFs of $(\boldsymbol{\tau}, \boldsymbol{u})$ summarizing the joint dynamics of the environment under joint policy $\boldsymbol{\pi}_{\bar{\boldsymbol{z}}}$. However, training centralized MAUSFs and using centralized GPI to achieve maximization over a combinatorial space of $\bar{\boldsymbol{z}}$ becomes impractical when there are more than a handful of agents, since the joint action

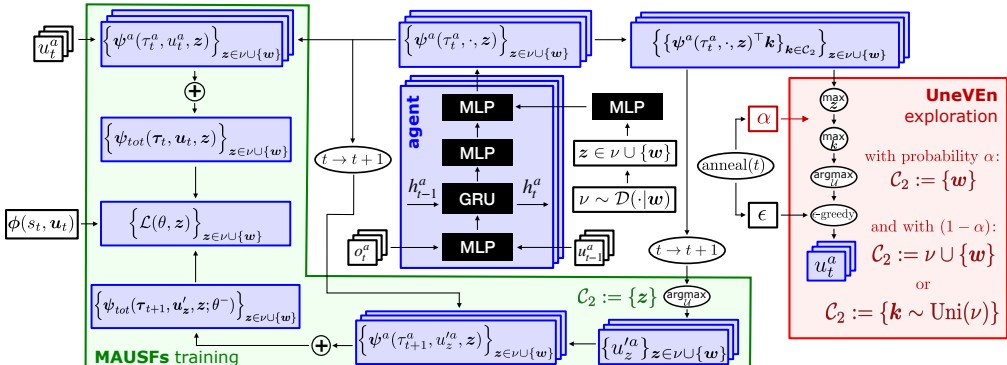

Figure 1: Schematic illustration of the MAUSFs training and UneVEn exploration with GPI policy.

space ($\mathcal{U}$) and joint task space ($\mathcal{C}^n$) of the agents grows exponentially with the number of agents. To leverage CTDE and enable decentralized execution by agents, we therefore propose novel *agent-specific SFs* for each agent $a$ following local policy $\pi^a_{\mathbf{z}^a}$, which condition only on its own local action-observation history and task $\mathbf{z}^a$.

**Decentralized Execution**: We define local *utility* functions for each agent $a$ as $Q^a(\tau^a, u^a, \mathbf{z}^a, \mathbf{w}) = \psi^a(\tau^a, u^a, \mathbf{z}^a; \theta)^\top \mathbf{w}$, where $\psi^a(\tau^a, u^a, \mathbf{z}^a; \theta)$ are the local agent-specific SFs induced by local policy $\pi^a_{\mathbf{z}^a}(u^a|\tau^a)$ of agent $a$ sharing parameters $\theta$. Intuitively, $Q^a(\tau^a, u^a, \mathbf{z}^a, \mathbf{w})$ is the utility function for agent $a$ when local policy $\pi^a_{\mathbf{z}^a}(u^a|\tau^a)$ of task $\mathbf{z}^a$ is executed on task $\mathbf{w}$. We use VDN decomposition to represent MAUSFs $\psi_{tot}$ as a sum of local agent-specific SFs for each agent $a$:

$$Q_{tot}(\boldsymbol{\tau}, \mathbf{u}, \bar{\mathbf{z}}, \mathbf{w}) = \sum_{a=1}^{n} Q^a(\tau^a, u^a, \mathbf{z}^a, \mathbf{w}) = \sum_{a=1}^{n} \psi^a(\tau^a, u^a, \mathbf{z}^a; \theta)^\top \mathbf{w} = \psi_{tot}(\boldsymbol{\tau}, \mathbf{u}, \bar{\mathbf{z}}; \theta)^\top \mathbf{w}. \quad (4)$$

We can now learn local agent-specific SFs $\psi^a$ for each agent $a$ that can be instantly weighted with any task vector $\mathbf{w} \in \mathbb{R}^d$ to generate local utility functions $Q^a$, thereby allowing agents to use the GPI policy in a decentralized manner.

**Decentralized Local GPI**: Our novel agent-specific SFs allows each agent $a$ to locally perform decentralized GPI by instant policy evaluation of a set $\mathcal{C}$ of local task policies $\{\pi^a_{\mathbf{z}^a}\}_{\mathbf{z}^a \in \mathcal{C}}$ on any unseen task $\mathbf{w}$ to compute a local GPI policy. Due to linearity of the VDN decomposition, this is equivalent to maximization over all combinations of $\bar{\mathbf{z}} \equiv \langle \mathbf{z}^1, \ldots, \mathbf{z}^n \rangle \in \mathcal{C} \times \ldots \times \mathcal{C} \equiv \mathcal{C}^n$ as:

$$\boldsymbol{\pi}'_{\mathbf{w}}(\boldsymbol{\tau}) \in \operatorname*{arg\,max}_{\mathbf{u} \in \mathcal{U}} \max_{\bar{\mathbf{z}} \in \mathcal{C}^n} Q_{tot}(\boldsymbol{\tau}, \mathbf{u}, \bar{\mathbf{z}}, \mathbf{w}) = \left\{ \operatorname*{arg\,max}_{u^a \in \mathcal{U}} \max_{\mathbf{z}^a \in \mathcal{C}} \psi^a(\tau^a, u^a, \mathbf{z}^a; \theta)^\top \mathbf{w} \right\}_{a=1}^{n}. \quad (5)$$

As all of the above relies on the linearity of the VDN decomposition, it cannot be directly applied to nonlinear mixing techniques like QMIX (Rashid et al., 2020b).

**Training**: MAUSFs for task combination $\bar{\mathbf{z}}$ are trained end-to-end by gradient descent on the loss:

$$\mathcal{L}(\theta, \bar{\mathbf{z}}) = \mathbb{E}_{\sim \mathcal{B}} \left[ \left\| \boldsymbol{\phi}(s_t, \mathbf{u}_t) + \gamma \psi_{tot}(\boldsymbol{\tau}_{t+1}, \mathbf{u}'_{\bar{\mathbf{z}}}, \bar{\mathbf{z}}; \theta^-) - \psi_{tot}(\boldsymbol{\tau}_t, \mathbf{u}_t, \bar{\mathbf{z}}; \theta) \right\|_2^2 \right], \quad (6)$$

where the expectation is over a minibatch of samples $\{(s_t, \mathbf{u}_t, \boldsymbol{\tau}_t)\}$ from the replay buffer $\mathcal{B}$ (Lin, 1992), $\theta^-$ denotes the parameters of a target network (Mnih et al., 2015) and joint actions $\mathbf{u}'_{\bar{\mathbf{z}}} = \{u'^a_{\mathbf{z}^a}\}_{a=1}^{n}$ are selected individually by each agent network using the current parameters $\theta$ (called Double $Q$-learning, van Hasselt et al., 2016): $u'^a_{\mathbf{z}^a} = \arg\max_{u \in \mathcal{U}} \psi^a(\tau^a_{t+1}, u, \mathbf{z}^a; \theta)^\top \mathbf{z}^a$. Each agent learns therefore local agent-specific SFs $\psi^a(\tau^a, u^a, \mathbf{z}; \theta)$ by gradient descent on $\mathcal{L}(\theta, \bar{\mathbf{z}})$ for all $\mathbf{z} \in \mathcal{C} \equiv \nu \cup \{\mathbf{w}\}$, where $\nu \sim \mathcal{D}(\cdot|\mathbf{w})$ is drawn from a distance measure around target task $\mathbf{w}$. The green region of Figure 1 shows a CTDE based architecture to train MAUSFs for a given target task $\mathbf{w}$. A detailed algorithm is present in Appendix A.

## 4 UNEVEN

In this section, we present UneVEn (red region of Figure 1), which leverages MAUSFs and decentralized GPI to enable efficient exploration on the target task $\mathbf{w}$. The joint-action value function of the target task $\mathbf{w}$ suffers from suboptimal approximations due to monotonic factorization. At

the beginning of every exploration episode, we sample a set of related tasks $\nu = \{z \sim \mathcal{D}(\cdot|w)\}$, containing potentially simpler reward functions, from a distribution $\mathcal{D}$ around the target task. The basic idea is that *some* of these related tasks can be efficiently learned using a monotonic joint-action value function. These tasks will therefore be solved early and exploration will concentrate on state-actions that are useful to them. As the sampled tasks are similar to $w$, this has the potential to put more weight on the *important joint actions* of the target task (Rashid et al., 2020a). This implicit weighting allows the learning of the joint-action value function of the target task to focus on accurately representing the value of the more important joint actions, and thereby overcome the relative overgeneralization pathology.

Many choices for $\mathcal{D}$ are possible, but in the following we sample related tasks using a normal distribution centered around the target task $w \in \mathbb{R}^d$ with a fixed variance $\sigma$ as $\mathcal{D} = \mathcal{N}(w, \sigma \mathbf{I}_d)$. The resulting task vectors weight the basis functions $\phi$ differently and represent different reward functions. In particular the varied reward functions can make these tasks much easier, but also harder, to solve with monotonic value functions. However, the approach has the advantage of not requiring any domain knowledge. The consequences of sampling harder tasks on learning are discussed with the corresponding action-selection schemes below.

**Action-Selection Schemes**: UneVEn uses two novel schemes to enable action selection based on related tasks. To emphasize the importance of the target task, we define a probability $\alpha$ of selecting actions based on the target task. Therefore, with probability $1-\alpha$, the action is selected based on the related task. Similar to other exploration schemes, $\alpha$ is annealed from 0.3 to 1.0 in our experiments over a fixed number of steps at the beginning of training. Once this exploration stage is finished (i.e., $\alpha = 1$), actions are always taken based on the target task's joint-action value function. Each action-selection scheme employs a local decentralized GPI policy, that maximizes over a set of policies $\pi_z$ based on $z \in \mathcal{C}_1$ (also referred to as the *evaluation* set) to estimate the $Q$-values of another set of tasks $k \in \mathcal{C}_2$ (also referred to as the *target* set) using:

$$u_t = \left\{ u_t^a = \arg\max_{u \in \mathcal{U}} \max_{k \in \mathcal{C}_2} \max_{z \in \mathcal{C}_1} \overbrace{\psi^a(\tau_t^a, u, z; \theta)^\top k}^{Q^a(\tau_t^a, u, z, k)} \right\}_{a \in \mathcal{A}}. \tag{7}$$

Here $\mathcal{C}_1 = \nu \cup \{w\}$ is the set of target and related tasks which induce the policies that are evaluated (dot-product) on the set of tasks $\mathcal{C}_2$, which varies with different action-selection schemes. The red box in Figure 1 illustrates UneVEn exploration. For example, $Q$-learning always picks actions based on the target task, i.e., the target set $\mathcal{C}_2 = \{w\}$. However, this scheme does not favour important joint actions. We call this default action-selection scheme **target GPI** and execute it with probability $\alpha$. We now propose two novel action-selection schemes based on related tasks with probability $1 - \alpha$, and thereby implicitly weighting joint actions during learning.

**Uniform GPI**: At the beginning of each episode, this action-selection scheme uniformly picks *one* related task, i.e., the target set $\mathcal{C}_2 = \{k \sim \text{Uniform}(\nu)\}$, and selects actions based on that task using the GPI policy throughout the episode. This uniform task selection explores the learned policies of all related tasks in $\mathcal{D}$. This works well in practice as there are often enough simpler tasks to induce the required bias over important joint actions. However, if the sampled related task is harder than the target task, the action-selection based on these harder tasks might hurt learning on the target task and lead to higher variance during training.

**Greedy GPI**: At every time-step $t$, this action-selection scheme picks the task $k \in \nu \cup \{w\}$ that gives the highest $Q$-value amongst the related and target tasks, i.e., the target set becomes $\mathcal{C}_2 = \nu \cup \{w\}$. Due to the greedy nature of this action-selection scheme, exploration is biased towards solved tasks, as those have larger values. We are thus exploring the solutions of tasks that are both solvable and similar to the target task $w$, which makes them great candidates for *important joint actions* of $w$.

**NO-GPI**: To demonstrate the influence of GPI on the above schemes, we also investigate ablations, where we define the evaluation set $\mathcal{C}_1 = \{k\}$ to only contain the currently estimated task $k$, i.e., using $u_t = \{u_t^a = \arg\max_{u \in \mathcal{U}} \max_{k \in \mathcal{C}_2} \psi^a(\tau_t^a, u, k; \theta)^\top k\}_{a \in \mathcal{A}}$ for action selection.

## 5 EXPERIMENTS

In this section, we evaluate UneVEn on a variety of complex domains. For evaluation, all experiments are carried out with five random seeds and results are shown with $\pm$ standard error across

seeds. We compare our method against a number of SOTA value-based MARL approaches: IQL (Tan, 1993), VDN (Sunehag et al., 2017), QMIX (Rashid et al., 2020b), MAVEN (Mahajan et al., 2019), WQMIX (Rashid et al., 2020a), QTRAN (Son et al., 2019), and QPLEX (Wang et al., 2020a).

**Domain 1 : $m$-step matrix game**

We first evaluate UneVEn on $m$-step matrix game proposed by Mahajan et al. (2019). This task is difficult to solve using simple $\epsilon$-greedy exploration policies as committed exploration is required to achieve the optimal return. Appendix E shows the $m$-step matrix game from Mahajan et al. (2019), in which the first joint decision of two agents determines the maximal outcome after another $m-1$ decisions. One initial joint action can reach a return of up to $m+3$, whereas another only allows for $m$. This challenges

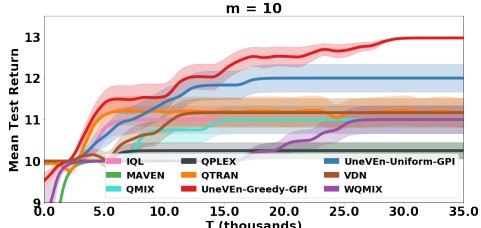

Figure 2: Baseline results for $m = 10$.

monotonic value functions, as the optimal joint reward function of the first decision is nonmonotonic. Figure 2 shows results of all methods on this task for $m = 10$ after training for $35k$ steps. UneVEn with greedy (UneVEn-Greedy-GPI) action selection scheme converges to an optimal return and both greedy and uniform (UneVEn-Uniform-GPI) schemes outperforms all other methods, which suffer from poor $\epsilon$-greedy exploration and often learn to take the suboptimal action in the beginning. Due to the nonmonotonicity of the initial state, it becomes difficult to switch the policy later, leading to suboptimal returns and only rarely converging to optimal solutions.

**Domain 2 : Cooperative Predator-Prey**

We next evaluate UneVEn on challenging cooperative predator-prey tasks similar to one proposed by Son et al. (2019), but significantly more complex in terms of the coordination required amongst agents. We use a complex partially observable predator-prey (PP) task involving eight agents (predators) and three prey that is designed to test coordination between agents, as each prey needs to be captured by at least three surrounding agents with a simultaneous *capture* action. If only one or two surrounding agents attempt to capture the prey, a negative reward of magnitude $p$ is given. Successful capture yields a positive reward of +1. This task is challenging for two reasons. First, depending on the magnitude of penalty $p$, exploration is difficult as even if a single agent miscoordinates, the penalty is given, and therefore, any steps toward successful coordination are penalized. Second, the agents must be able to differentiate between the values of successful and unsuccessful collaborative actions, which monotonic value functions can only do if all agents already act optimally. More details about the task are available in Appendix B.

**Proposition 1.** For the predator-prey game defined above, the optimal joint action reward function for any group of $2 \leq k \leq n$ predator agents surrounding a prey is *nonmonotonic* (as defined by Mahajan et al., 2019) iff $p > 0$. (Proof is provided in Appendix B).

**Simpler PP Tasks**: We first demonstrate that both VDN and QMIX with monotonic joint-action value functions can learn on target tasks with simpler reward functions. To generate a simpler task, we remove the penalty associated with miscoordination, i.e., $p = 0$, thereby making the returns monotonic. Figure 3 shows that both QMIX and VDN can solve this task as there is no miscoordination penalty and the monotonic joint-action value function can learn to efficiently represent the optimal joint-action val-

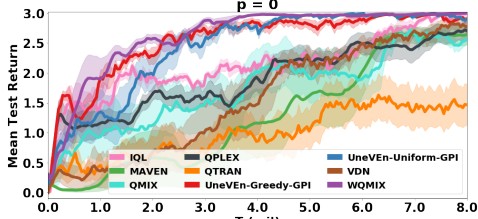

Figure 3: Baseline results for $p = 0$.

ues. Other SOTA value-based approaches (MAVEN, WQMIX and QPLEX) and UneVEn with both uniform (UneVEn-Uniform-GPI) and greedy (UneVEn-Greedy-GPI) action-selection schemes can also solve this monotonic target task.

**Harder PP Tasks**: We now make the target task nonmonotonic by increasing the magnitude of the penalty associated with *each* miscoordination, i.e., $p \in \{0.004, 0.008, 0.012, 0.016\}$. For a smaller penalty of $p = 0.004$, Figure 4 (top left) shows that VDN is still able to solve the task, further suggesting that simpler reward related tasks (with lower penalties) can be solved with monotonic approximations. However, both QMIX and VDN fail to learn on three other higher penalty target

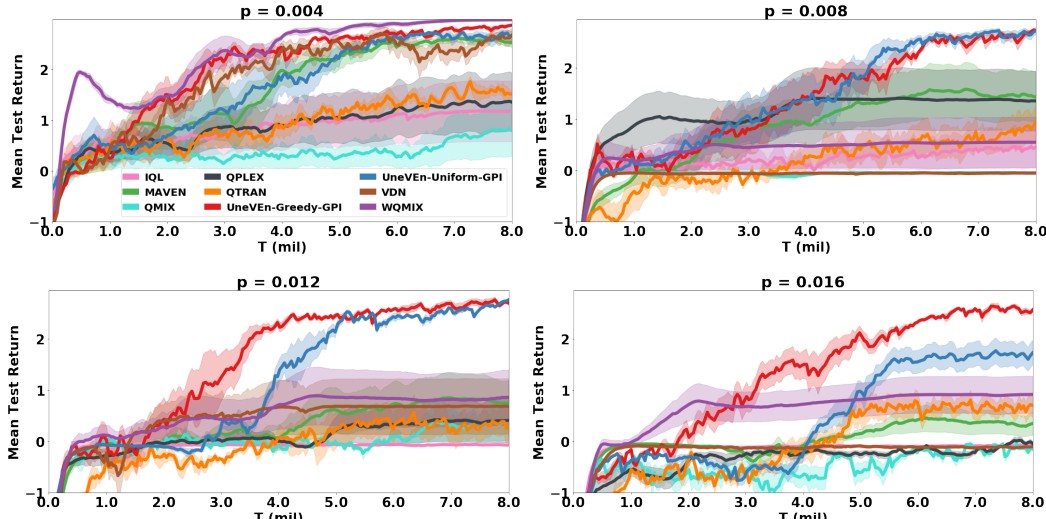

Figure 4: Comparison between UneVEn and SOTA MARL baselines with $p \in \{0.004, 0.008, 0.012, 0.016\}$.

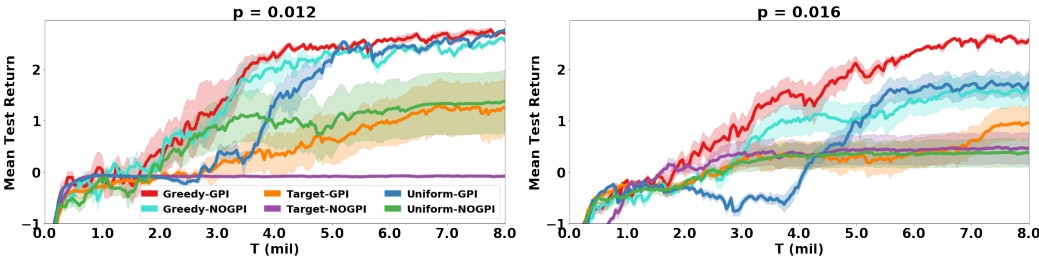

Figure 5: Ablation results: Comparison between different action selection of UneVEn for $p \in \{0.012, 0.016\}$.

tasks due to their monotonic constraints, which hinder the accurate learning of the joint-action value functions. Intuitively, when uncoordinated joint actions are much more likely than coordinated ones, the penalty term can dominate the average value estimated by each agent's utility. This makes it difficult to learn an accurate monotonic approximation that will select the optimal joint actions.

Interestingly, other SOTA value-based approaches that aim to address the monotonicity restriction of QMIX and VDN such as MAVEN, QTRAN, WQMIX and QPLEX also fail to learn on higher penalty tasks. WQMIX solves the task when $p = 0.004$, but fails on other three higher penalty target tasks. Although WQMIX uses an explicit weighting mechanism to bias learning towards important joint actions, it must identify these actions by learning a nonmonotonic value function first. An $\epsilon$-greedy exploration based on the target task will take a long time to learn such a value function, which is visible in the large standard error for $p \in \{0.008, 0.012, 0.016\}$ in Figure 4. By contrast, both UneVEn-Uniform-GPI and UneVEn-Greedy-GPI can approximate nonmonotonic value functions more accurately and solve the task for all values of $p$. As expected, the variance of UneVEn-Uniform-GPI is high on higher penalty target tasks (for e.g., $p = 0.016$) as exploration suffers from action selection based on harder related tasks. UneVEn-Greedy-GPI does not suffer from this problem. Videos of learnt policies are available at `https://rb.gy/rdwpo5`.

**Ablations**: Figure 5 shows ablation results for higher penalty tasks, i.e., $p = \{0.012, 0.016\}$. To contrast the effect of UneVEn on exploration, we compare our two novel action-selection schemes to UneVEn-Target-GPI, which only selects the greedy actions of the target task. The results clearly show that UneVEn-Target-GPI fails to solve the higher penalty nonmonotonic tasks as the employed monotonic joint value function of the target task fails to accurately represent the values of different joint actions. This demonstrates the critical role of UneVEn and its action-selection schemes.

Next we evaluate the effect of GPI by comparing against UneVEn with MAUSFs without using the GPI policy, i.e., setting the evaluation set $\mathcal{C}_1 = \{k\}$ in Equation 7. First, UneVEn using a NOGPI policy with both uniform (Uniform-NOGPI) and greedy (Greedy-NOGPI) action selection outperform Target-NOGPI, further strengthening the claim that UneVEn with its novel action-selection

scheme enables efficient exploration and bias towards optimal joint actions. Next, Figure 5 clearly shows that for each corresponding action-selection scheme (uniform, greedy, and target), using a GPI policy (∗-GPI) is always favourable as it performs either similarly to the NOGPI policy (∗-NOGPI) or much better. GPI appears to improve zero-shot generalization of MAUSFs across tasks, which in turn enables good action selection for related tasks during UneVEn exploration.

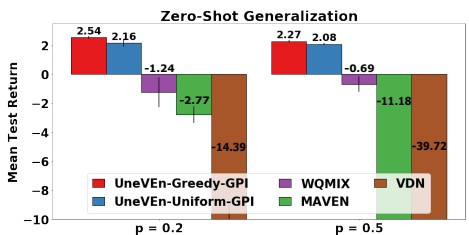

**Zero-Shot Generalization**: Lastly, we evaluate this zero-shot generalization for all methods to check if the learnt policies are useful for unseen high penalty test tasks. We train all methods for 8 million environmental steps on a task with $p = 0.004$, and test 60 rollouts of the resulting policies of all methods that are able to solve the training task, i.e., UneVEn-Greedy-GPI, UneVEn-Uniform-GPI, VDN, MAVEN, and WQMIX, on tasks with $p \in \{0.2, 0.5\}$. For policies trained with UneVEn-Greedy-GPI and UneVEn-Uniform-GPI, we use the NOGPI policy for

Figure 6: Zero-shot generalization comparison; training on $p = 0.004$, testing on $p \in \{0.2, 0.5\}$.

the zero-shot testing, i.e., $\mathcal{C}_1 = \mathcal{C}_2 = \{w\}$. Figure 6 shows that UneVEn with both uniform and greedy schemes exhibits great zero-shot generalization and solves both test tasks even with very high penalties. As MAUSFs learn the reward's basis functions, rather than the reward itself, zero-shot generalization to larger penalties follow naturally. Furthermore, using UneVEn exploration allows the agents to collect enough diverse behaviour to come up with a near optimal policy for the test tasks. On the other hand, the learnt policies for all other methods that solve the target task with $p = 0.004$ are ineffective in these higher penalty nonmonotonic tasks, as they do not learn to avoid unsuccessful capture attempts. More details about the implementations are included in Appendix C. Additional ablation experiments are discussed in Appendix D.

### Domain 3 : Starcraft Multi-Agent Challenge (SMAC)

We now evaluate UneVEn on challenging cooperative StarCraft II maps from the SMAC benchmark (Samvelyan et al., 2019). We consider SMAC maps where each ally agent unit is additionally penalized for being killed or suffering damage from the enemy, in addition to receiving positive reward for killing/inflicting damage on enemy units, which has recently shown to improve performance (Son et al., 2020). We present the results for one super hard map (`MMM2`, involving 10 units of 3 types), two hard asymmetric maps (`5m_vs_6m` and `10m_vs_11m`) and three easy maps (`2s3z`, `1c3s5z` and `8m`).

Figure 7 presents the mean test win rate for all maps. Both VDN and QMIX achieve almost 100% win rate on these maps, which leads us to conclude that they do not suffer from relative overgeneralization and that simple $\epsilon$-greedy policies suffices for these maps. Thus, the additional complexity of learning MAUSFs in our approach results in slightly slower convergence. However, UneVEn with both GPI schemes matches the performance as VDN and QMIX in most maps, with only small deviations in `5m_vs_6m`, demonstrating that our method can scale well to large complex tasks.

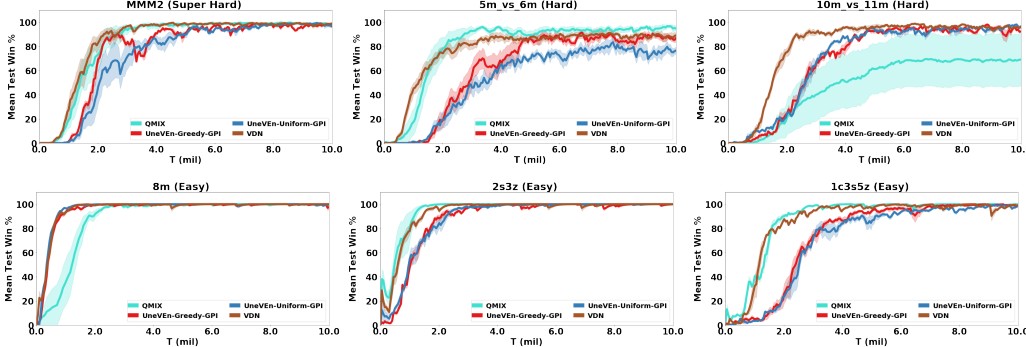

Figure 7: Comparison between UneVEn, VDN and QMIX on SMAC maps.

## 6 RELATED WORK

**Improving monotonic value function factorization in CTDE** MAVEN (Mahajan et al., 2019) shows that the monotonic joint-action value function of QMIX and VDN suffers from suboptimal approximations on nonmonotonic tasks. It addresses this problem by learning a diverse ensemble of monotonic joint-action value functions conditioned on a latent variable by optimizing the mutual information between the joint trajectory and the latent variable. Deep Coordination Graphs (DCG) (Böhmer et al., 2020) uses a predefined coordination graph (Guestrin et al., 2002) to represent the joint-action value function. However, DCG is not a fully decentralized approach and specifying the coordination graph can require significant domain knowledge. Son et al. (2019) propose QTRAN that addresses the monotonic restriction of QMIX by learning a (decentralizable) VDN-factored joint-action value function along with an unrestricted centralized critic. The corresponding utility functions are distilled from the critic by solving a linear optimization problem involving all joint actions, but its exact implementation is computationally intractable and the corresponding approximate versions have instable performance. QPLEX (Wang et al., 2020a) uses a duplex dueling (Wang et al., 2016) network architecture to factorize the joint-action value function with linear decomposition structure. WQMIX (Rashid et al., 2020a) learns a QMIX-factored joint-action value function along with an unrestricted centralized critic and proposes explicit weighting mechanisms to bias the monotonic approximation of the optimal joint-action value function towards important joint actions, which is similar to our work. However, in our work, the weightings are implicitly done through action-selection based on simpler reward related tasks, which are easier to solve.

**Exploration** There exists a plethora of techniques for exploration in model-free single-agent RL, based on intrinsic novelty reward (Bellemare et al., 2016; Tang et al., 2017), predictability (Pathak et al., 2017), pure curiosity (Burda et al., 2019) or Bayesian posteriors (Osband et al., 2016; Gal et al., 2017; Fortunato et al., 2018; O'Donoghue et al., 2018). In the context of multi-agent RL, Böhmer et al. (2019) discuss the influence of unreliable intrinsic reward and Wang et al. (2020b) quantify the influence that agents have on each other's return. Zheng & Yue (2018) propose to coordinate exploration between agents by shared latent variables, whereas Jaques et al. (2018) investigate social motivations of competitive agents. However, these techniques aim to visit as much of the state-action space as possible, which exacerbates the relative overgeneralization pathology. Approaches that use state abstraction (e.g., Roderick et al., 2018) can speed up exploration, but only by restricting the considered space with prior knowledge. In contrast, UneVEn explores similar *tasks*. This guides exploration to states and actions that prove *useful*, which restricts the explored space and overcomes relative overgeneralization. To the best of our knowledge, the only other work that explores the task space is Leibo et al. (2019): they use the evolution of competing agents as an auto-curriculum of harder and harder tasks. Collaborative agents cannot compete against each other, though, and their approach does therefore not affect relative overgeneralization.

**Successor Features** Most of the work on SFs have been focused on single-agent settings (Dayan, 1993; Kulkarni et al., 2016; Lehnert et al., 2017; Zhu et al., 2017; Barreto et al., 2017; 2018; Borsa et al., 2018; Lehnert & Littman, 2019; Lee et al., 2019; Hansen et al., 2019) for transfer learning and zero-shot generalization across tasks with different reward functions. Gupta et al. (2019) *uses* single-agent SFs in a transition-independent multi-agent setting to estimate the probability of events.

## 7 CONCLUSION

This paper presents novel multi-agent universal successor features (MAUSFs) decomposed as local agent-specific SFs, which enables decentralized version of the GPI to maximize over a combinatorial space of agent policies, making MAUSFs a perfect fit for MARL. We then propose UneVEn, which leverages the generalization power of MAUSFs to perform action-selection based on simpler related tasks to address the issue of sub-optimality of target task's monotonic joint-action value function in current SOTA methods. Our experiments show that UneVEn significantly outperforms VDN, QMIX and other state-of-the-art value-based MARL methods on nonmonotonic tasks by a substantial margin.

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

## A  TRAINING ALGORITHM

Algorithm 1 presents the training of MAUSFs with UneVEn. Our method is able to learn on all tasks (target $\boldsymbol{w}$ and sampled $\boldsymbol{z}$) simultaneously in a sample efficient manner using the same feature $\boldsymbol{\phi}_t \equiv \boldsymbol{\phi}(s_t, \boldsymbol{u}_t)$ due to the linearly decomposed reward function (Equation 1).

---

**Algorithm 1** Training MAUSFs with UneVEn

---

**Require:** $\epsilon, \alpha, \beta$ target task $\boldsymbol{w}$, set of agents $\mathcal{A}$, standard deviation $\sigma$

1: **procedure** TRAIN:
2:     Initialize the local-agent SF network $\boldsymbol{\psi}^a(\tau^a, u^a, \boldsymbol{z}; \theta)$ and replay buffer $\mathcal{M}$
3:     **for** fixed number of **epochs do**
4:         $\nu \sim \mathcal{N}(\boldsymbol{w}, \sigma \mathbf{I}_d)$;   $\boldsymbol{o}_0 \leftarrow$ RESETENV()                        $\triangleright$ $\boldsymbol{o}_t \equiv \{o_t^a\}_{a \in \mathcal{A}}$
5:         $t = 0$;   $\mathcal{M} \leftarrow$ NEWEPISODE$(\mathcal{M}, \nu, \boldsymbol{o}_0)$
6:         **while** not **terminated do**
7:             **if** Bernoulli$(\epsilon)$=1 **then** $\boldsymbol{u}_t \leftarrow$ Uniform$(\mathcal{U})$
8:             **else** $\boldsymbol{u}_t \leftarrow$ UNEVEN$(\boldsymbol{\tau}_t, \nu)$
9:             $\langle \boldsymbol{o}_{t+1}, \boldsymbol{\phi}_t \rangle \leftarrow$ ENVSTEP$(\boldsymbol{u}_t)$
10:            $\mathcal{M} \leftarrow$ ADDTRANSITION$(\mathcal{M}, \boldsymbol{u}_t, \boldsymbol{o}_{t+1}, \boldsymbol{\phi}_t)$
11:            $t \leftarrow t + 1$
12:         $\mathcal{L} \leftarrow 0$;   $\mathcal{B} \leftarrow$ SAMPLEMINIBATCH$(\mathcal{M})$
13:         **foreach** $\{\boldsymbol{\tau}_t, \boldsymbol{u}_t, \boldsymbol{\phi}_t, \boldsymbol{\tau}_{t+1}, \nu\} \in \mathcal{B}$ **do**
14:             **foreach** $\boldsymbol{z} \in \nu \cup \{\boldsymbol{w}\}$ **do**
15:                 $\boldsymbol{u}_{\boldsymbol{z}}' \leftarrow \big\{ \arg\max_{u \in \mathcal{U}} \boldsymbol{\psi}^a(\tau_{t+1}^a, u, \boldsymbol{z}; \theta)^\top \boldsymbol{z} \big\}_{a \in \mathcal{A}}$
16:                 $\mathcal{L} \leftarrow \mathcal{L} + \big\| \boldsymbol{\phi}_t + \gamma \boldsymbol{\psi}_{tot}(\boldsymbol{\tau}_{t+1}, \boldsymbol{u}_{\boldsymbol{z}}', \boldsymbol{z}; \theta^-) - \boldsymbol{\psi}_{tot}(\boldsymbol{\tau}_t, \boldsymbol{u}_t, \boldsymbol{z}; \theta) \big\|_2^2$
17:         $\theta \leftarrow$ OPTIMIZE$(\theta, \nabla_\theta \mathcal{L})$
18:         $\theta^- \leftarrow (1 - \beta)\,\theta^- + \beta\,\theta$
19: **procedure** UNEVEN$(\boldsymbol{\tau}_t, \nu)$:
20:     **if** Bernoulli$(\alpha) = 1$ **or** Scheme is **Target then**
21:         $\mathcal{C}_2 \leftarrow \{\boldsymbol{w}\}$
22:     **else**
23:         **if** Scheme is **Uniform then**
24:             $\mathcal{C}_2 \leftarrow \nu \sim$ Uniform$(\nu)$
25:         **else if** Scheme is **Greedy then**
26:             $\mathcal{C}_2 \leftarrow \nu \cup \{\boldsymbol{w}\}$
27:     **if** Use_GPI_Policy is **True then**
28:         $\mathcal{C}_1 \leftarrow \nu \cup \{\boldsymbol{w}\}$
29:         $\boldsymbol{u}_t \leftarrow \{u_t^a = \arg\max_{u \in \mathcal{U}} \max_{\boldsymbol{k} \in \mathcal{C}_2} \max_{\boldsymbol{z} \in \mathcal{C}_1} \boldsymbol{\psi}^a(\tau_t^a, u, \boldsymbol{z}; \theta)^\top \boldsymbol{k}\}_{a \in \mathcal{A}}$
30:     **else**
31:         $\boldsymbol{u}_t \leftarrow \{u_t^a = \arg\max_{u \in \mathcal{U}} \max_{\boldsymbol{k} \in \mathcal{C}_2} \boldsymbol{\psi}^a(\tau_t^a, u, \boldsymbol{k}; \theta)^\top \boldsymbol{k}\}_{a \in \mathcal{A}}$
        **return** $\boldsymbol{u}_t$

---

## B  EXPERIMENTAL DOMAIN DETAILS AND ANALYSIS

We consider a complicated partially observable predator-prey (PP) task in an $10 \times 10$ grid involving eight agents (predators) and three preys that is designed to test coordination between agents, as each prey needs a simultaneous *capture* action by at least three surrounding agents to be captured. Each agent can take 6 actions i.e. move in one of the 4 directions (Up, Left, Down, Right), remain still (no-op), or try to catch (capture) any adjacent prey. The prey moves around in the grid with a probability of 0.7 and remains still at its position with probability 0.3. Impossible actions for both agents and prey are marked unavailable, for eg. moving into an occupied cell or trying to take a capture action with no adjacent prey.

| | A1-Capture | | | A1-Other | |
| --- | --- | --- | --- | --- | --- |
| | A2-Capture | A2-Other | | A2-Capture | A2-Other |
| A3-Capture | +1 | $-p$ | A3-Capture | $-p$ | $-p$ |
| A3-Other | $-p$ | $-p$ | A3-Other | $-p$ | 0 |

Table 1: Joint-Reward function of three agents surrounding a prey. The first table indicates joint-rewards when Agent 1 takes capture action and second table indicates joint-rewards when Agent 1 takes any other action. Notice that there are numerous joint actions leading to penalty $p$.

If either a single or a pair of agents take a capture action on an adjacent prey, a negative reward of magnitude $p$ is given. If three or more agents take the capture action on an adjacent prey, it leads to a successful capture of that prey and yield a positive reward of $+1$. The maximum possible reward for capturing all preys is therefore $+3$. Each agent observes a $5 \times 5$ grid centered around its position which contains information showing other agents and preys relative to its position. An episode ends if all preys have been captured or after $800$ time steps. This task is similar to one proposed by Böhmer et al. (2020); Son et al. (2019), but significantly more complex in terms of the coordination required amongst agents as more agents need to coordinate simultaneously to capture the preys. We now prove Proposition 1 which states that:

**Proposition.** For, the predator-prey game defined above, the optimal joint action reward function for any group of $2 \leq k \leq n$ predator agents surrounding a prey is *nonmonotonic* (as defined by Mahajan et al., 2019) iff $p > 0$.

*Proof.* Without loss of generality, we assume a single prey surrounded by three agents $(A_1, A_2, A_3)$ in the environment. The joint reward function for this group of three agents is defined in Table 1.

For the case $p > 0$ the proposition can be easily verified using the definition of non-monotonicity (Mahajan et al., 2019). For any $3 \leq k \leq n$ agents attempting to catch a prey in state $s$, we fix the actions of any $k - 3$ agents to be "*other*" indicating either of up, down, left, right, and noop actions and represent it with $\boldsymbol{u}^{k-3}$. Next we consider the rewards $r$ for two cases:

- If we fix the action of any *two* of the remaining three agents as "other" represented as $\boldsymbol{u}^2$, the action of the remaining agent becomes $u^1 = \arg\max_{u \in \mathcal{U}} r(s, \langle u, \boldsymbol{u}^2, \boldsymbol{u}^{k-3} \rangle) = $ "other".

- If we fix the $\boldsymbol{u}^2$ to be "capture", we have : $u_1 = \arg\max_{u \in \mathcal{U}} r(s, \langle u, \boldsymbol{u}^2, \boldsymbol{u}^{k-3} \rangle) = $ "capture".

Thus the best action for agent $A_1$ in state $s$ depends on the actions taken by the other agents and the rewards $R(s)$ are non-monotonic. Finally for the equivalence, we note that for the case $p = 0$ we have that a default action of "capture" is always optimal for any group of $k$ predators surrounding the prey. Thus the rewards are monotonic as the best action for any agent is independent of the rest. □

## C  IMPLEMENTATION DETAILS

### C.1  HYPER PARAMETERS

All algorithms are implemented in the PyMARL framework (Samvelyan et al., 2019). All our experiments use $\epsilon$-greedy scheme where $\epsilon$ is decayed from $\epsilon = 1$ to $\epsilon = 0.05$ over $250k$ time steps. All our tasks use a discount factor of $\gamma = 0.99$. We freeze the trained policy every $30k$ timesteps and run 20 evaluation episodes with $\epsilon = 0$. We use learning rate of $0.0005$ with soft target updates for all experiments. We use a target network similar to Mnih et al. (2015) with "soft" target updates, rather than directly copying the weights: $\theta^- \leftarrow \beta * \theta + (1 - \beta) * \theta^-$, where $\theta$ are the current network parameters. We use $\beta = 0.005$ for all experiments. This means that the target values are constrained to change slowly, greatly improving the stability of learning. All algorithms were trained with RMSprop optimizer by one gradient step on loss computed on a batch of 32 episodes sampled from a replay buffer containing last 1000 episodes. We also used gradient clipping to restrict the norm of the gradient to be $\leq 10$.

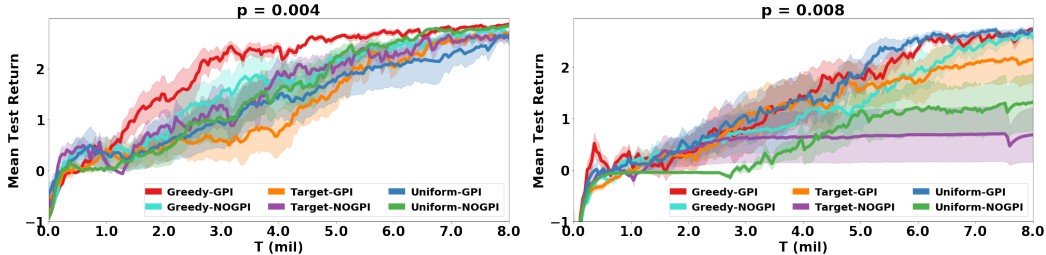

Figure 8: Additional Ablation results: Comparison between different action selection of UneVEn for $p \in \{0.004, 0.008\}$.

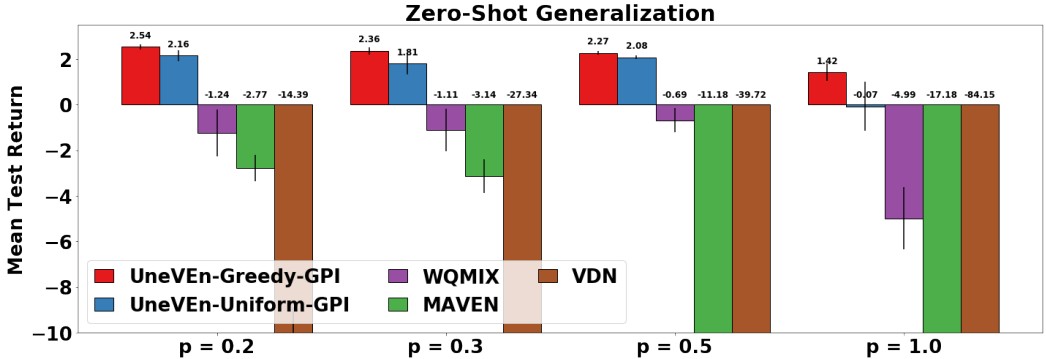

Figure 9: Additional Zero-shot generalization results for $p \in \{0.2, 0.3, 0.5, 1.0\}$.

The probability $\alpha$ of action selection based on target task in UneVEn with uniform and greedy action selection schemes increases from $\alpha = 0.3$ to $\alpha = 1.0$ over $250k$ time steps. For sampling related tasks using normal distribution, we use $\mathcal{N}(\boldsymbol{w}, \sigma\mathbf{I}_d)$ centered around target task $\boldsymbol{w}$ with $\sigma \in \{0.1, 0.2\}$. At the beginning of each episode, we sample *six* related tasks, therefore $|\nu| = 6$.

## C.2 NN ARCHITECTURE

Each agent's local observation $o_t^a$ are concatenated with agent's last action $u_{t-1}^a$, and then passed through a fully-connected (FC) layers of 128 neurons, followed by ReLU activation, a GRU (Chung et al., 2014), and another FC of the same dimensionality to generate a action-observation history summary for the agent. Each agent's task vector $\boldsymbol{z} \in \nu \cup \{\boldsymbol{w}\}$ is passed through a FC layer of 128 neurons followed by ReLU activation to generate an internal task embedding. The history and task embedding are concatenated together and passed through two hidden FC-256 layers and ReLU activations to generate the outputs for each action. For methods with non-linear mixing such as QMIX (Rashid et al., 2020b), WQMIX (Rashid et al., 2020a), and MAVEN (Mahajan et al., 2019), we adopt the same hypernetworks from the original paper and test with either a single or double hypernet layers of dim 64 utilizing an ELU non-linearity. For all baseline methods, we use the code shared publicly by the corresponding authors on Github.

## D ADDITIONAL RESULTS

Figure 8 presents additional ablation results for comparison between UneVEn with different action selection schemes for $p \in \{0.004, 0.008\}$. Figure 9 presents additional zero-shot generalization results for policies trained on target task with penalty $p = 0.004$ tested on tasks with penalty $p \in \{0.2, 0.3, 0.5, 1.0\}$. For UneVEn-Greedy-GPI, we can observe that the average number of miscoordinated capture attempts per episode actually drops with $p$ and converges around 1.5, i.e., for return $R_p$, average mistakes per episode is $\frac{3-R_p}{p} = \{2.3, 2.1, 1.5, 1.6\}$ for $p \in \{0.2, 0.3, 0.5, 1.0\}$.

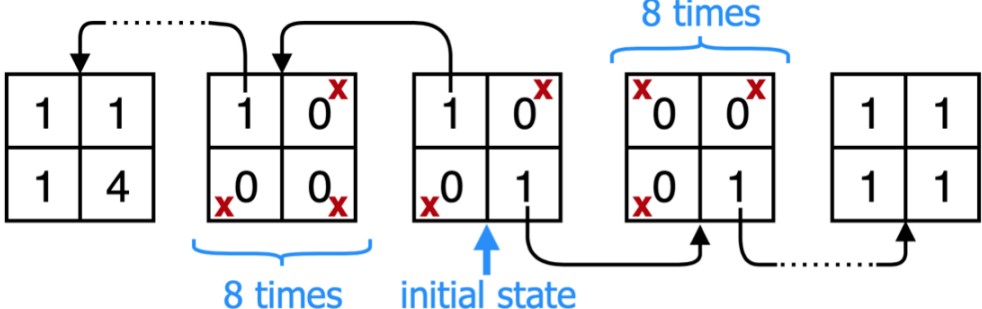

Figure 10: $m$-step matrix game from Mahajan et al. (2019) for $m = 10$. The red cross means that selecting that joint action will lead to termination of the episode.

## E $m$-STEP MATRIX GAMES

Figure 10 shows the $m$-step matrix game for $m = 10$ from Mahajan et al. (2019), where there are $m - 2$ intermediate steps, and selecting a joint-action with zero reward leads to termination of the episode.

