# OpenReview forum: "UneVEn: Universal Value Exploration for Multi-Agent Reinforcement Learning"
_ICLR.cc/2021/Conference — Reject_

### Official Review · AnonReviewer2 · 2020-10-28
**clever idea, limited experiments**

**Rating:** 5
**Confidence:** 2

**Review:**

The paper develops and evaluates an algorithm for decision making in the CTDE MARL setting (centralized training and decentralized execution for multiagent reinforcement learning). That is, the concern is how to use closely supervised training to produce agents that can work independently toward a common goal. The problem is formalized in the DEC-POMDP (decentralized partially observable Markov decision process) setting.

The proposed solution uses elements of several recent and promising approaches including value decomposition networks, QMIX (a nonlinear monotonic combination of individual utility functions), linear reward functions, universal value functions, successor features, generalized policy improvement. The novel idea is combining these ideas to create multi-agent universal successor features.

The power of the resulting algorithm was demonstrated in a set of predator-prey games with increased difficulty of learning to coordinate due to an increasing penalty value for accidental miscoordination. It was shown that other methods for this problem were not able to handle learning in the setting of high costs for coordination, but the proposed algorithm could.

The paper is quite good for what it claims to do. But, my impression is that the contribution is not very large. Specifically, the main demonstration was (significantly) improved performance on a predator-prey task. However, I am skeptical that these results will generalize  to other domains---these kinds of problems are quite hard and it's not clear if ANY algorithm would be expected to solve a diverse collection of real-world challenges. In the context of what the paper was trying to do, additional domains (3 distinct ones, say?) would go a long way toward characterizing the space of domains where this approach is appropriate.

I guess I'm saying that this algorithm might be THE algorithm you'd want to use (it creatively combines a number of promising component ideas), but the paper didn't provide an argument for why the reader should be convinced that the positive results generalize.

Some detailed comments:

"We now propose two novel action-selection schemes based on related tasks with probability 1-eps, and thereby implicitly weighting joint actions during learning." I'm afraid I wasn't able to parse this sentence. Reword?

pg. 6: Maybe putting the graphs on the same y-axis would aid in comparisons between settings (different values of p).

"However, both QMIX and VDN fail to learn on other three higher penalty target tasks" -> "However, both QMIX and VDN fail to learn on three other higher penalty target tasks"?

"between different action selection" -> "between action-selection schemes".

"learns a QMIX-factored joint-action value function along with an unrestricted centralized critic and propose" -> "learns a QMIX-factored joint-action value function along with an unrestricted centralized critic and proposes"?

---

> ### Author Response · Authors · 2020-11-20
> **Official Response to Reviewer2**
>
> We thank the reviewer for their valuable feedback. We have added additional results on two more domains in the paper: (a) Section 5 Domain 1 : a non-monotonic m-step matrix game from [1] to test how non-monotonicity and exploration interact, and (b) Section 5 Domain 3 : Starcraft Micromanagement Challenge (SMAC) from [2].
>
> References:
>
> [1] Mahajan, A., Rashid, T., Samvelyan, M. and Whiteson, S., 2019. Maven: Multi-agent variational exploration. In Advances in Neural Information Processing Systems (pp. 7613-7624).
>
> [2] Samvelyan, M., Rashid, T., de Witt, C.S., Farquhar, G., Nardelli, N., Rudner, T.G., Hung, C.M., Torr, P.H., Foerster, J. and Whiteson, S., 2019. The starcraft multi-agent challenge. arXiv preprint arXiv:1902.04043.

---

### Official Review · AnonReviewer3 · 2020-10-28
**Not well motivated and lack of significance**

**Rating:** 3
**Confidence:** 5

**Review:**

This paper employs the linear value factorization proposed by VDN and extends universal successor features with GPI to multi-agent reinforcement learning. It also proposes an exploration method, called universal value exploration, that biases an agent’s action selection by utilizing related tasks. Empirical evaluation demonstrates its outperformance over baselines in predator-prey tasks.

However, I have some major concerns about this paper.  First, the motivation of this paper is confusing, that is, the monotonic restriction leads to inefficient exploration. Exploration and the expressiveness of value factorization do not have a causal relation. Some exploration strategies can allow VDN and QMIX to learn the optimal strategy even with non-monotonic returns. One of such trivial exploration policies is the optimal (or nearly optimal) policy.

The proposed MAUSFs approach looks like a simple extension of single-agent successor features to multi-agent settings by using linear value factorization.  Are there challenges for such an extension?

I don’t think the proposed UNEVEN exploration method can address the relative overgeneralization pathology. The relative overgeneralization pathology is caused by the limited function class of some value factorization methods. QTRAN and QPLEX can potentially address this issue, but they may not perform well in some tasks that require more efficient exploration.

I don’t think it is fair to compare UNEVEN with most of the baselines in this paper. It is because UNEVEN uses the linear parameterized representation of reward functions that allows generating related tasks. In contrast, some baselines do not aim to address the exploration problem or do not take additional information.

The experimental evaluation can be improved. For example, it can include more complex settings, like SMAC.

---

> ### Author Response · Authors · 2020-11-20
> **Official Response to Reviewer3 (Part 1)**
>
> We thank the reviewer for their valuable feedback.
>
> **Q1: "First, the motivation of this paper is confusing, that is, the monotonic restriction leads to inefficient exploration. Exploration and the expressiveness of value factorization do not have a causal relation."**
>
> A1: MAVEN [1] (Theorem 1 and 2) show that VDN and QMIX agents (with both uniform and epsilon greedy policies) can latch onto suboptimal behaviour early on during learning, due to the monotonicity constraint as it can prevent the network from correctly remembering the true value of the optimal action (currently perceived as suboptimal). These suboptimal approximations in turn affect exploration as suboptimal actions keep getting selected based on maximizing the current suboptimal joint action value function. WQMIX [2] shows that non-monotonic tasks can be solved with monotonic value functions (like VDN and QMIX) if the optimal actions are sampled more often during training. There is, therefore, a causal relationship between how actions are explored (e.g. with $\epsilon$-greedy) and which tasks can be learned by monotonic factorizations like VDN and QMIX.
>
> As explained in our related works section, most work on exploration in RL focuses on visiting as much of the state-action space as possible which in turn exacerbates the relative overgeneralization pathology. Therefore, in our case, we sample simpler reward related tasks with similar optimal joint actions as the target task, thereby implicitly biasing the learning towards important joint-actions.
>
> ---
>
> **Q2: "Some exploration strategies can allow VDN and QMIX to learn the optimal strategy even with non-monotonic returns. One of such trivial exploration policies is the optimal (or nearly optimal) policy."**
>
> A2: The reviewer correctly points out that putting all "weight" on the optimal joint actions during learning (i.e. only sample optimal joint actions) should allow us to learn any non-monotonic task (if we ignore function approximation errors). The obvious issue is that we don’t know that exploration policy. Therefore, we propose an exploration approach which samples and learns tasks similar to the target task $\vec{w}$ in the hope that their optimal actions are often useful for the target task, too. Our empirical results justify this hope and show that our exploration approach can alleviate the problem of joint action value function trying to represent all joint actions, leading to suboptimal approximations and poor exploration. We don’t see how an optimal (or nearly) optimal policy is a “trivial” exploration policy.
>
> ---
>
> **Q3: "The proposed MAUSFs approach looks like a simple extension of single-agent successor features to multi-agent settings by using linear value factorization.  Are there challenges for such an extension?"**
>
> A3: The VDN decomposition of centralized USFs is straightforward, however, the insight that our local GPI implicitly performs global GPI over all combinations of sampled agent-policies is novel. The major contribution of the paper is the UneVEn exploration scheme that allows to overcome the monotonic restrictions of a VDN factorization in a way that would be impossible without local SFs. Our approach is novel as follows:
>
> 1. Our approach enables decentralized execution by introducing novel agent-specific SFs while taking advantage of centralized training (CTDE).
> 2. Our approach enables decentralized GPI which is particularly well suited for MARL, as it allows us to maximize over a combinatorial set of agent policies.
> 3. We introduce novel action-selection schemes by leveraging VDN-factorized MAUSFs to overcome the representation limitation of VDN/QMIX.
> 4. Our results along with ablations show that our approach significantly outperforms other methods on non monotonic tasks and shows comparable performance on other large scale tasks.
>
> ---

---

> > ### Author Response · Authors · 2020-11-20
> > **Official Response to Reviewer3 (Part 2)**
> >
> > **Q4: "I don’t think the proposed UNEVEN exploration method can address the relative overgeneralization pathology. The relative overgeneralization pathology is caused by the limited function class of some value factorization methods. QTRAN and QPLEX can potentially address this issue, but they may not perform well in some tasks that require more efficient exploration."**
> >
> > A4: As we have argued above, relative overgeneralization can be overcome by sampling the optimal actions of the target task more often. For this purpose, the exploration in action space must be modified. QTRAN, QPLEX and WQMIX do this with epsilon-greedy sampling from an unconstrained value function. However, our results demonstrate that our method, which samples the optimal actions of similar monotonic tasks, performs this exploration significantly better.
> >
> > ---
> >
> > **Q5: "I don’t think it is fair to compare UNEVEN with most of the baselines in this paper. It is because UNEVEN uses the linear parameterized representation of reward functions that allows generating related tasks. In contrast, some baselines do not aim to address the exploration problem or do not take additional information."**
> >
> > A5: In this paper, our task is to show that epsilon-greedy policies used by most current SOTA methods like QPLEX, QTRAN, WQMIX even with enhanced representational capacities fail to learn on nonmonotonic tasks due to suboptimal approximations of the values of different (coordinated and uncoordinated) joint actions. Allowing to generate related tasks does not make our comparison unfair as our goal is to solve the harder target task. There is a lot of work on curriculum learning that shows learning on an entire curriculum (allowing generating a sequence of source tasks) is better than learning the target task from scratch. Some baselines like VDN, QMIX, IQL do not aim to address the exploration problem which is a key point we are trying to showcase as well in the paper.
> >
> > ---
> >
> > **Q6: "The experimental evaluation can be improved. For example, it can include more complex settings, like SMAC."**
> >
> > A6: We have added additional results on two more domains in the paper: (a) Section 5 Domain 1 : a non-monotonic m-step matrix game from [1] to test how nonmonotonicity and exploration interact, and (b) Section 5 Domain 3 : starcraft micromanagement challenge (SMAC) from [4].
> >
> > ---
> >
> > References:
> >
> > [1] Mahajan, A., Rashid, T., Samvelyan, M. and Whiteson, S., 2019. Maven: Multi-agent variational exploration. In Advances in Neural Information Processing Systems (pp. 7613-7624).
> >
> > [2] Rashid, T., Farquhar, G., Peng, B. and Whiteson, S., 2020. Weighted QMIX: Expanding Monotonic Value Function Factorisation for Deep Multi-Agent Reinforcement Learning. Advances in Neural Information Processing Systems, 33.
> >
> > [3] Son, K., Kim, D., Kang, W.J., Hostallero, D.E. and Yi, Y., 2019. Qtran: Learning to factorize with transformation for cooperative multi-agent reinforcement learning. arXiv preprint arXiv:1905.05408.
> >
> > [4] Samvelyan, M., Rashid, T., de Witt, C.S., Farquhar, G., Nardelli, N., Rudner, T.G., Hung, C.M., Torr, P.H., Foerster, J. and Whiteson, S., 2019. The starcraft multi-agent challenge. arXiv preprint arXiv:1902.04043.

---

> > > ### Comment · AnonReviewer3 · 2020-11-24
> > > **Concerns not addressed yet**
> > >
> > > Thanks for your response. However, my concerns have not been addressed yet.
> > >
> > > 1.	Every RL or MARL algorithm contains the following fundamental components: Generalization (i.e., specifying a function class), Exploration (i.e., generating data), Loss function (i.e., optimization objective), and Optimization (i.e., finding optimal or good policies). Because they all affect the solution, they are correlated, but not causal.
> > > 2.	For Q2, the “trivial” example is to show that the non-monotonic return cannot stop VDN and QMIX from learning the optimal strategy.  In some sense, WQMIX takes the idea of this “trivial” example, which learns an “optimal” joint value function and uses it as a reference point to guide exploration.
> > > 3.	I appreciate that the authors agreed that the proposed MAUSFs is a quite straightforward extension of single-agent successor features. I don’t think this is a problem if the proposed method can be extensively evaluated and shown its promise. However, I am not quite satisfied with the evaluation conducted in this paper, as discussed below.
> > > 4.	In general, experiments were designed to favor the proposed method, which cannot help to evaluate the generality of the proposed method. I suggested the authors evaluate in SMAC. I appreciate their efforts to conduct such experiments, but they changed the original benchmark settings, which makes the proposed method work. In addition, the proposed method still underperforms VDN and QMIX even after engineering the benchmark. I suggested the authors conduct a more extensive evaluation on the SMAC benchmark and also compared with state-of-the-art methods in this setting, which also aims to address the monotonicity limitation of VDN and QMIX, including MAVEN, QPLEX, and WQMIX.
> > >
> > > All in all, the direction this paper takes is interesting and I encourage the authors to further improve it. But the current version is not ready yet. I will lower my score.

---

> > > > ### Author Response · Authors · 2020-11-24
> > > > **Thanks for your comments.**
> > > >
> > > > We thank the reviewer for their comments.
> > > >
> > > > **For Q2, the “trivial” example is to show that the non-monotonic return cannot stop VDN and QMIX from learning the optimal strategy.  In some sense, WQMIX takes the idea of this “trivial” example, which learns an “optimal” joint value function and uses it as a reference point to guide exploration.**
> > > >
> > > > While WQMIX can learn non-monotonic tasks given the optimal Q-value Q*, the algorithm must still learn Q* using $\epsilon$-greedy exploration. As our experiments show, this can take a long time and may in some cases never succeed due to function approximation. UneVEn provides better exploration by solving a range of related tasks and, by choosing the tasks with the highest value, executing the optimal policies of those which are monotonic. This is not a guarantee for optimal exploration, but we believe our empirical results, in particular in the matrix game and predator-prey domain that is designed to test exploration, justify our approach.
> > > >
> > > > ---
> > > >
> > > > **In general, experiments were designed to favor the proposed method, which cannot help to evaluate the generality of the proposed method. I suggested the authors evaluate in SMAC. I appreciate their efforts to conduct such experiments, but they changed the original benchmark settings, which makes the proposed method work. In addition, the proposed method still underperforms VDN and QMIX even after engineering the benchmark. I suggested the authors conduct a more extensive evaluation on the SMAC benchmark and also compared with state-of-the-art methods in this setting, which also aims to address the monotonicity limitation of VDN and QMIX, including MAVEN, QPLEX, and WQMIX.**
> > > >
> > > > We use negative rewards (penalizing agents for getting killed) for our SMAC results as it has shown to improve performance in recent work [1]. Using negative rewards actually leads to better performance of VDN and QMIX over original benchmark results [2]. We only experiment with VDN and QMIX for SMAC as VDN, QMIX and our method are able to solve SMAC maps with close to 100% win rates. We will perform more extensive evaluation on the SMAC benchmark.
> > > >
> > > > References:
> > > >
> > > > [1] Son et al 2020. QTRAN++: Improved Value Transformation for Cooperative Multi-Agent Reinforcement Learning. arXiv preprint arXiv:2006.12010.
> > > >
> > > > [2] Rashid, T., Samvelyan, M., De Witt, C.S., Farquhar, G., Foerster, J. and Whiteson, S., 2020. Monotonic Value Function Factorisation for Deep Multi-Agent Reinforcement Learning. arXiv preprint arXiv:2003.08839.

---

### Official Review · AnonReviewer4 · 2020-10-28

**Rating:** 6
**Confidence:** 5

**Review:**

##########################################################################
Summary:

This paper studies the problem setting of cooperative multi-agent RL (coop-MARL) under centralized training with decentralized execution (CTDE). Additionally, it assumes that the reward function is known and is represented as a weighted linear combination of a basis function.
They consider Q-learning with value-decomposition as the basis for tackling such problems and improve upon SOTA methods for coop-MARL in the domain of predator-prey (PP). Specifically, they claim to improve upon monotonic value-decomposition methods such as VDN and QMIX by better addressing the issue of relative overgeneralization which stems from their monotonicity constraint in representing the joint-action value function. This paper has a similar objective as Weighted QMIX, but it achieves it through implicit weighting by learning over a set of related tasks (reward function weights).
Their approach is to extend Value-Decomposition Networks (VDN) by combining it with a multi-agent version of USFs (Successor Features + reward-function-based Universal Value Functions), called MAUSFs. Then, they use a related-task sampler during exploration to sample simpler tasks and through this, they claim that more importance is placed on better joint actions implicitly.

##########################################################################
Reasons for score:

Overall, I think this is, in general, a sound paper and so would like to see it accepted. As a paper to inspire and give insights on issues in coop-MARL through a creative approach, I like this paper. However, regarding the general applicability and its practical usefulness, I'm not fully convinced yet. Additionally, I have some other concerns which, hopefully, the authors can address during the rebuttal period.

##########################################################################
Pros:

- The approach is innovative.

- The experiments are targeted and illustrative. The zero-generalization experiment is also useful.

- Paper is well-written and covers the literature appropriately.

##########################################################################
Cons:

Please see my questions below.

##########################################################################
Questions during the rebuttal period:

1) How can we ensure pi_z is optimal on task z? Isn't the optimality of pi_z on task z based on which the SF is learned a necessary condition for the GPI update to apply?

2) I'm not fully convinced how in general this method can be expected to improve regarding the issue of relative overgeneralization? How do we know that these related tasks would bias the monotonic approximation towards "more important" joint actions? Is there a guarantee in the limit of tasks sampled by some distribution D?

3) I have some concerns regarding the general applicability of this approach. To the best of my understanding, this work is limited to domains where the reward function is known and represented as a linear combination of some basis function. To me, this seems like a very challenging issue to overcome as breaking down the reward function based on experienced reward signals is perhaps as hard as learning the optimal action-value function (loosely speaking).
So can this approach really lead to more generic extensions that would work on unknown and general reward functions?

4) Why does VDN's gap w.r.t. UneVEn narrows down (instead of widening) in p=0.004 (nonmonotonic) vs. p=0 (monotonic)?

5) Why does QMIX perform so much worse than VDN in p=0.004 (in fact, on all tasks in Figures 2 & 3 this is the case)? Shouldn't QMIX's ability to represent a larger class of decompositions improve performance over VDN?

##########################################################################
Minor comments:

- Section 1, paragraph 3: "QTRAN (Son et al., 2019) and WQMIX (Rashid et al., 2020a) addresses..." -> address

- Remove the first comma in Proposition 1: "For, the predator-prey game defined above,..."

- I think \pi*_z should be used to indicate the optimal policy on task z is meant.

---

> ### Author Response · Authors · 2020-11-20
> **Official Response to Reviewer4 (Part 1)**
>
> We thank the reviewer for their valuable feedback.
>
> **Q1: "How can we ensure pi_z is optimal on task z? Isn't the optimality of pi_z on task z based on which the SF is learned a necessary condition for the GPI update to apply?"**
>
> A1: Our method samples tasks related to the target task based on a normal distribution with a predefined variance at the beginning of each episode and performs TD-error updates to each of the related tasks and the target task. This ensures that the learning happens for all sampled related tasks.
>
> Using Proposition 1 from [1] : Given a GPI policy computed for task $\vec{w}$ based on a set $\nu$ of related tasks, the maximum difference between optimal policy for task $\vec{w}$ and the GPI policy for any state-action pair is upper bounded by two major terms: (a) minimum of the norm of distance between task $\vec{w}$ and any task $\vec{z} \in \nu$,  (b) the approximation error of SFs of task $\vec{z}$.
>
> In our case, the first error is quite small as we always sample related tasks in close vicinity of the target task with small variance of 0.1 or 0.2. The second error is taken care of by learning simultaneously on all related tasks.
>
> ---
>
> **Q2: "I'm not fully convinced how in general this method can be expected to improve regarding the issue of relative overgeneralization? How do we know that these related tasks would bias the monotonic approximation towards "more important" joint actions? Is there a guarantee in the limit of tasks sampled by some distribution D?"**
>
> A2: Relative overgeneralization in nonmonotonic MARL tasks happens when the number of suboptimal joint actions leading to negative rewards is much greater than the number of optimal joint actions and the employed joint action value function has limited representational capacity [2, 3]. MAVEN [2] (Theorem 1 and 2) show that VDN and QMIX agents (with both uniform and epsilon greedy policies) can latch onto suboptimal behaviour early on during learning, due to the monotonicity constraint as it can prevent the network from correctly remembering the true value of the optimal action (currently perceived as suboptimal). These suboptimal approximations in turn affect exploration as suboptimal actions keep getting selected based on maximizing the current suboptimal joint action value function.
>
> WQMIX [4] shows that non-monotonic tasks can be solved with monotonic value functions (like VDN and QMIX) if the optimal actions are sampled more often during training. They introduce explicit weighting mechanisms to enable bias towards such important joint actions during learning through an unconstrained critic. We rely on the intuition that executing similar, but simpler tasks solvable by monotonic value functions, changes the action distribution in our favour. For example, while solving the task for higher p = 0.016, UneVEn could potentially sample tasks with lower penalties like p = 0.004, 0.003, etc., which are easier to solve using a monotonic value function and have similar important joint actions (see VDN in Figure 4a with p = 0.004). We achieve this by sampling tasks similar to the target task at random and executing the task with the largest Q-value (Greedy-GPI action selection scheme). This increases the chance of sampling the target task's optimal actions and therefore makes it solvable with a VDN factorization. The Uniform-GPI scheme suffers from action-selection based on harder related tasks as well resulting in more variance during learning (see Figure 4d with p = 0.016).
>
> The ablations in the updated version of the paper (Figure 5) which always picks actions based on the target task i.e. Target-GPI and Target-NOGPI clearly show the importance of action selection based on related tasks as they are unable to solve the tasks. We do not have any theoretical guarantees in the limit of sampled related tasks.
>
> ---

---

> > ### Author Response · Authors · 2020-11-20
> > **Official Response to Reviewer4 (Part 2)**
> >
> > **Q3: "I have some concerns regarding the general applicability of this approach. To the best of my understanding, this work is limited to domains where the reward function is known and represented as a linear combination of some basis function. To me, this seems like a very challenging issue to overcome as breaking down the reward function based on experienced reward signals is perhaps as hard as learning the optimal action-value function (loosely speaking). So can this approach really lead to more generic extensions that would work on unknown and general reward functions?"**
> >
> > A3: The reviewer correctly points out that in this paper we rely on the assumption that both the basis functions $\phi$ and the reward-weights $\vec{w}$ of the target task are known. However, [5] shows that both can be learned from a given set of reward functions, which span the intended space of basis functions, using multitask regression. The question of how one could learn a set of bases that allow effective UneVEn exploration from experienced rewards alone, is very interesting, but out of the scope of this paper and we leave it as future work.
> >
> > ---
> >
> > **Q4: "Why does VDN's gap w.r.t. UneVEn narrows down (instead of widening) in p=0.004 (nonmonotonic) vs. p=0 (monotonic)?"**
> >
> > A4: We can only explain this observation with the variance over the 5 seeds we tested. Other baselines like WQMIX also show different behaviour in the beginning of training here. However, a variety of tested algorithms converged to a similar final performance for both p=0 and p=0.004, which leads us to believe that a punishment of 0.004 is close enough to a monotonic task to be learnable by monotonic value functions. None of these algorithms learn the tasks with larger punishments, though, which emphasizes the importance of UneVEn exploration in strongly non-monotonic tasks.
> >
> > ---
> >
> > **Q5: "Why does QMIX perform so much worse than VDN in p=0.004 (in fact, on all tasks in Figures 2 & 3 this is the case)? Shouldn't QMIX's ability to represent a larger class of decompositions improve performance over VDN?"**
> >
> > A5: QMIX does indeed represent a larger class of monotonic value functions compared to VDN, however, its additional representational complexity over VDN takes longer to learn and leads to slower convergence even in the case of p = 0.0, and fails to converge on any higher penalty tasks including p = 0.004. Similar results have been shown by [3, 6, 7] where VDN and QMIX converge to a similar performance. [7] also shows that VDN performs better than QMIX in some Starcraft SMAC maps.
> >
> > ---
> >
> > **Additional Comment: "To be specific, the experiments study only the game of predator-prey and do not yet demonstrate the algorithm’s generalization power beyond this motivating task. It should outperform existing approaches on a wider range of tasks where the monotonicity does not hold, and be at least competitive on tasks where such an assumption hold, to be valuable."**
> >
> > Response: We have also added additional results on two more domains in the paper: (a) Section 5 Domain 1 : a non-monotonic m-step matrix game from [2] to test how nonmonotonicity and exploration interact, and (b) Section 5 Domain 3 : starcraft micromanagement challenge (SMAC) from [8].
> >
> > ---
> >
> > References:
> >
> > [1] Borsa, D., Barreto, A., Quan, J., Mankowitz, D., Munos, R., van Hasselt, H., Silver, D. and Schaul, T., 2018. Universal successor features approximators. arXiv preprint arXiv:1812.07626.
> >
> > [2] Mahajan, A., Rashid, T., Samvelyan, M. and Whiteson, S., 2019. Maven: Multi-agent variational exploration. In Advances in Neural Information Processing Systems (pp. 7613-7624).
> >
> > [3]  Wendelin Böhmer, Vitaly Kurin, and Shimon Whiteson. Deep coordination graphs. In Proceedings of Machine Learning and Systems (ICML), pp. 2611–2622, 2020.
> >
> > [4] Rashid, T., Farquhar, G., Peng, B. and Whiteson, S., 2020. Weighted QMIX: Expanding Monotonic Value Function Factorisation for Deep Multi-Agent Reinforcement Learning. Advances in Neural Information Processing Systems, 33.
> >
> > [5] Barreto, A., Hou, S., Borsa, D., Silver, D. and Precup, D., 2020. Fast reinforcement learning with generalized policy updates. Proceedings of the National Academy of Sciences.
> >
> > [6] Son, K., Kim, D., Kang, W.J., Hostallero, D.E. and Yi, Y., 2019. Qtran: Learning to factorize with transformation for cooperative multi-agent reinforcement learning. arXiv preprint arXiv:1905.05408.
> >
> > [7] Rashid, T., Samvelyan, M., De Witt, C.S., Farquhar, G., Foerster, J. and Whiteson, S., 2020. Monotonic Value Function Factorisation for Deep Multi-Agent Reinforcement Learning. arXiv preprint arXiv:2003.08839.
> >
> > [8] Samvelyan, M., Rashid, T., de Witt, C.S., Farquhar, G., Nardelli, N., Rudner, T.G., Hung, C.M., Torr, P.H., Foerster, J. and Whiteson, S., 2019. The starcraft multi-agent challenge. arXiv preprint arXiv:1902.04043.

---

> > > ### Comment · AnonReviewer4 · 2020-11-23
> > > **Thanks for your detailed response!**
> > >
> > > Thanks for your detailed response!
> > >
> > > Your answers to Qs 1-2 & 5 helped me verify that I've not made a mistake in my original assessment. Regarding Q2, I think the connection between how your approach handles specifically the problem of relative overgeneralization is not strong. This may be intuitive to think that your method can, in the discussed manner, handle relative overgeneralization, but beyond giving an intuition the paper should not make a general statement.
> > >
> > > The answer to Q3 clarified how this method could more generally be applicable in certain problem settings. This is nice but makes your approach specific to scenarios where reward functions can be learned first. I'm sure that there are application scenarios where this approach would be a good fit, but when you think how much additional assumptions need to be in place for this method to outperform a simple VDN, QMIX or even the most basic (yet most general) IQL, it then starts to seem less appealing as a way forward. Nevertheless, I still like the paper for giving us a new viewpoint on handling the issue of relative overgeneralization.
> > >
> > > I find the answer to Q4 unsatisfactory: for a paper that at submission time only showed results on a simple predator-prey toy problem, I find it insufficient to only use 5 seeds, which as you stated, constrains analysis of the results about the specific properties of the considered methods (**"We can only explain this observation with the variance over the 5 seeds we tested, ..."**). For a paper which mainly relies on intuition and empirical results to support said intuitions, this is simply insufficient to me.
> > >
> > > All in all, I thank the authors for their response, but I'm not able to raise my score beyond the original score of 6 at this point. However, I will remain open to this during the next discussion phase.

---

### Official Review · AnonReviewer1 · 2020-10-29
**There are still some issues**

**Rating:** 5
**Confidence:** 4

**Review:**

Some popular methods like VDN and QMIX focus on the monotonic factorization of joint-action value function, which is not realistic in non-monotonic cases when the agent’s best action depends on other agents’ actions. This phenomenon is common. For example, in the prisoner’s dilemma the value function can be monotonically decreasing in each of the agent’s local value function. One of the effect this paper focuses on is that the monotonically factorization lacks the representational capacity to distinguish the values of coordinated and uncoordinated joint actions during exploration. This effect is well explained in the predator-prey game example, where both VDN and QMIX have undesired performance.

Recent work like QTRAN and WQMIX tried to address the problem of inefficient exploration caused by monotonic factorization. However, these approaches still rely on inefficient-greedy exploration which may fail on harder tasks e.g., again, the predator-prey task above with higher value of p.

This paper applies universal successor features to the multi-agent setting (Multi-Agent Universal Successor Features MAUSFs). Decentralized agent-specific SFs with VDN enables agents to compute decentralized greedy policies and to perform decentralized local GP. The two components have some good synergy and improves VDN to some extent. The propose Universal Value Exploration (UneVEn) can solve tasks with nonmonotonic values. Both the combination of SF and VDN and the UneVEn algorithm are very intuitive and are easy to understand and implement.

However, the strength of the results are moderate. As it is very natural to combine the two it suffices to figure out how strong the synergy is between. This is not observed either through theoretical insights or experimental studies. To be specific, the experiments study only the game of predator-prey and do not yet demonstrate the algorithm’s generalization power beyond this motivating task. It should outperform existing approaches on a wider range of tasks where the monotonicity does not hold, and be at least competitive on tasks where such an assumption hold, to be valuable. The experiment on Zero-shot generalization is interesting, but SF along should be able to present such an effect of transfer learning (generalization). The paper needs to show how the synergy between SF and VDN is really like in the task of transfer.

Pro
 	1. This paper keeps up with the frontier of MARL. It discusses a lot of algorithms proposed recently and provides a detailed background knowledge.

2. It clearly stated the current research gap in value function factorization.

Weak points
 	1. This paper might not fill the research gap it mentioned very well. For example, in the introduction part, it says the shortages of VDN and QMIX is that they are restricted by monotonic property. However, it states in the UneVEn section that, "the basic idea is some of the related tasks can be efficiently learned using a monotonic joint-action value function. " Does it mean that the implementation of UneVEn still relies on monotonic value function?

2. The author can elaborate more on the experiments from an intuitive view, rather than just stating the experiment data.
 	For example, as the experiment shows, the UneVEn only outperforms as the penalty parameter p get larger and larger. What's the reason behind it? How does UneVEn overcome the "curse" of suboptimal stuck in VDN and QMIX?

3. More experiments on general tasks should be provided.

---

> ### Author Response · Authors · 2020-11-20
> **Official Response to Reviewer1**
>
> We thank the reviewer for their valuable feedback.
>
> **Q1: "This paper might not fill the research gap it mentioned very well. For example, in the introduction part, it says the shortages of VDN and QMIX is that they are restricted by monotonic property. However, it states in the UneVEn section that, “the basic idea is some of the related tasks can be efficiently learned using a monotonic joint-action value function.” Does it mean that the implementation of UneVEn still relies on monotonic value function?"**
>
> A1: WQMIX [1] shows that non-monotonic tasks can be solved with monotonic value functions (like VDN and QMIX) if the optimal actions are sampled more often during training. They introduce explicit weighting mechanisms to enable bias towards such important joint actions during learning through an unconstrained critic. We rely on the intuition that executing similar, but simpler tasks solvable by monotonic value functions, changes the action distribution in our favour. For example, while solving the task for higher p = 0.016, UneVEn could potentially sample tasks with lower penalties like p = 0.004, 0.003, etc., which are easier to solve using a monotonic value function and have similar important joint-actions (see VDN in Figure 4a, p = 0.004). We achieve this by sampling tasks similar to the target task at random and executing the task with the largest Q-value (Greedy-GPI action selection scheme). This increases the chance of sampling the target task's optimal actions and therefore makes it solvable with a VDN factorization.
>
> ---
>
> **Q2: "The author can elaborate more on the experiments from an intuitive view, rather than just stating the experiment data. For example, as the experiment shows, the UneVEn only outperforms as the penalty parameter p gets larger and larger. What's the reason behind it? How does UneVEn overcome the "curse" of suboptimal stuck in VDN and QMIX?"**
>
> A2: As shown in the reward matrix (Table 1 in Appendix B) for predator prey, there are numerous joint actions which lead to a penalty and only a single joint action which leads to a successful capture (reward +1). For VDN and QMIX with simple $\epsilon$-greedy policies, as we increase the penalty, the TD squared loss computed from states in which the prey is surrounded gets dominated by the actions which lead to a penalty (that are more numerous than the optimal actions). This makes it difficult to learn an accurate monotonic approximation that can capture the optimal joint action [2]. UneVEn tackles this issue by efficient exploration based on related tasks with simpler reward functions but with similar important joint actions.
>
> For example, while solving the task for higher p = 0.016, UneVEn could potentially sample tasks with lower penalties like p = 0.004, 0.003, etc., which can be solved by monotonic value functions and have similar important joint-actions. UneVEn with Greedy-GPI always picks the task with the highest Q-value which should refer to simpler reward related tasks which are solved earlier and have higher Q-values than other higher penalty related tasks. UneVEn with Uniform-GPI might suffer from action-selection based on higher penalty related tasks leading to higher variance during learning (see Figure 4d, p = 0.016).
>
> ---
>
> **Q3: "More experiments on general tasks should be provided."**
>
> A3: We have added additional results on two more domains in the paper: (a) Section 5 Domain 1 : a non-monotonic m-step matrix game from [2] to test how non-monotonicity and exploration interact, and (b) Section 5 Domain 3 : Starcraft Micromanagement Challenge (SMAC) from [3].
>
> ---
>
> References:
>
> [1] Rashid, T., Farquhar, G., Peng, B. and Whiteson, S., 2020. Weighted QMIX: Expanding Monotonic Value Function Factorisation for Deep Multi-Agent Reinforcement Learning. Advances in Neural Information Processing Systems, 33.
>
> [2] Mahajan, A., Rashid, T., Samvelyan, M. and Whiteson, S., 2019. Maven: Multi-agent variational exploration. In Advances in Neural Information Processing Systems (pp. 7613-7624).
>
> [3] Samvelyan, M., Rashid, T., de Witt, C.S., Farquhar, G., Nardelli, N., Rudner, T.G., Hung, C.M., Torr, P.H., Foerster, J. and Whiteson, S., 2019. The starcraft multi-agent challenge. arXiv preprint arXiv:1902.04043.

---

> > ### Comment · AnonReviewer1 · 2020-11-23
> > **Experiments**
> >
> > The additional experiments in SMAC seem to demonstrate the competitiveness of Uneven, but it might be more relevant if the authors can show the effectiveness of Uneven by showing the outperformance in at least some domain other than predator-prey. This manuscript can then be much more valuable for its publicity.

---

> > > ### Author Response · Authors · 2020-11-23
> > > **Thanks for your comments.**
> > >
> > > In addition to showing competitive performance of UneVEn on SMAC maps, we have also added results on a non-monotonic m-step game (Section 5 Domain 1) from [1] where UneVEn outperform all other methods significantly.
> > >
> > > [1] Mahajan, A., Rashid, T., Samvelyan, M. and Whiteson, S., 2019. Maven: Multi-agent variational exploration. In Advances in Neural Information Processing Systems (pp. 7613-7624).

---

### Decision · Program_Chairs · 2021-01-07
**Final Decision**

**Decision:**

Reject

**Comment:**

This paper adapts the ideas around universal successor features for decentralised multi-agent environments, with a particular emphasis on deriving better exploration from them. Like most of the reviewers, I think this is indeed a promising research direction. Given the complexity of the endeavour however, it may take a few more steps until the empirical evidence can back up the authors' ambition: the reviewers' consensus on the current version of the paper is that it is not ready for publication yet.